# Multiple blood feeding in mosquitoes shortens the *Plasmodium falciparum* incubation period and increases malaria transmission potential

W. Robert Shaw[1][º], Inga E. Holmdahl[2,3][º], Maurice A. Itoe[1], Kristine Werling[1], Meghan Marquette[1], Douglas G. Paton[1], Naresh Singh[1], Caroline O. Buckee[2,3], Lauren M. Childs[4]*, Flaminia Catteruccia[1]*

1 Department of Immunology and Infectious Diseases, Harvard T.H. Chan School of Public Health, Boston, Massachusetts, United States of America, 2 Department of Epidemiology, Harvard T.H. Chan School of Public Health, Boston, Massachusetts, United States of America, 3 Center for Communicable Disease Dynamics, Harvard T.H. Chan School of Public Health, Boston, Massachusetts, United States of America, 4 Department of Mathematics, Virginia Tech, Blacksburg, Virginia, United States of America

º These authors contributed equally to this work.
* lchilds@vt.edu (LMC); fcatter@hsph.harvard.edu (FC)

**Data Availability Statement:** All relevant data are within the manuscript and its Supporting Information files.

## Abstract

Many mosquito species, including the major malaria vector *Anopheles gambiae*, naturally undergo multiple reproductive cycles of blood feeding, egg development and egg laying in their lifespan. Such complex mosquito behavior is regularly overlooked when mosquitoes are experimentally infected with malaria parasites, limiting our ability to accurately describe potential effects on transmission. Here, we examine how *Plasmodium falciparum* development and transmission potential is impacted when infected mosquitoes feed an additional time. We measured *P. falciparum* oocyst size and performed sporozoite time course analyses to determine the parasite's extrinsic incubation period (EIP), i.e. the time required by parasites to reach infectious sporozoite stages, in *An. gambiae* females blood fed either once or twice. An additional blood feed at 3 days post infection drastically accelerates oocyst growth rates, causing earlier sporozoite accumulation in the salivary glands, thereby shortening the EIP (reduction of 2.3 ± 0.4 days). Moreover, parasite growth is further accelerated in transgenic mosquitoes with reduced reproductive capacity, which mimic genetic modifications currently proposed in population suppression gene drives. We incorporate our shortened EIP values into a measure of transmission potential, the basic reproduction number $R_0$, and find the average $R_0$ is higher (range: 10.1%–12.1% increase) across sub-Saharan Africa than when using traditional EIP measurements. These data suggest that malaria elimination may be substantially more challenging and that younger mosquitoes or those with reduced reproductive ability may provide a larger contribution to infection than currently believed. Our findings have profound implications for current and future mosquito control interventions.

**Funding:** F.C. is funded by a Faculty Research Scholar Award by the Howard Hughes Medical Institute (HHMI) and the Bill & Melinda Gates Foundation (BMGF) (Grant ID: OPP1158190, www. hhmi.org), and by the National Institutes of Health (NIH) (R01 AI124165, R01 AI153404, www.nih. gov). L.M.C. is supported by the National Science Foundation (NSF) (Grant ID: 1853495, www.nsf. gov). The findings and conclusions within this publication are those of the authors and do not necessarily reflect positions or policies of the HHMI, the BMGF, or the NIH. The funders had no role in the study design, in data collection, analysis or interpretation, in the decision to publish, or the preparation of the manuscript.

**Competing interests:** The authors have declared that no competing interests exist.

## Author summary

In natural settings the female *Anopheles gambiae* mosquito, the major malaria vector, blood feeds multiple times in her lifespan. Here we demonstrate that an additional blood feed accelerates the growth of *Plasmodium falciparum* malaria parasites in this mosquito. Incorporating these data into a mathematical model across sub-Saharan Africa reveals that malaria transmission potential is likely to be higher than previously thought, making disease elimination more difficult. Additionally, we show that control strategies that manipulate mosquito reproduction with the aim of suppressing *Anopheles* populations may inadvertently favor malaria transmission. Our data also suggest that parasites can be transmitted by younger mosquitoes, which are less susceptible to insecticide killing, with negative implications for the success of insecticide-based strategies.

## Introduction

Malaria remains a devastating disease for tropical and subtropical regions, accounting for an estimated 405,000 deaths and 228 million cases in 2018 [1]. *Anopheles* mosquitoes transmit the causative *Plasmodium* malaria parasites, and malaria control strategies aimed at the mosquito vector through long-lasting insecticide-treated bed nets (LLINs) and indoor residual spraying (IRS) have greatly decreased the malaria burden in recent decades. Despite this progress, a deeper understanding of the impact of complex mosquito behaviors on the transmission potential of mosquito populations is needed to generate better predictions of disease transmission and of the efficacy of control interventions. This need is all the more urgent given that the reduction in malaria cases has plateaued in the past few years [1], possibly due to the reduced efficacy of LLINs and IRS in the face of insecticide resistance spreading in mosquitoes [2].

A key determinant of malaria transmission is the significant length of time it takes for *Plasmodium* parasites to develop from sexual stages in the mosquito blood meal into infectious sporozoites in the salivary glands, a time period known as the extrinsic incubation period or EIP. As mosquito survival must exceed the EIP for onward transmission to humans to occur, this parameter has an important relationship to mosquito mortality. This is one reason why life-shortening insecticidal interventions like LLINs emerge as particularly effective in epidemiological models (in addition to lowering mosquito densities and biting rates [3–5]).

In most epidemiological studies the *Plasmodium falciparum* EIP is considered to last 12–14 days [6–8]. This is similar to the expected lifespan of the *Anopheles* female, which, although difficult to measure precisely due to the lack of reliable age markers, has been shown to be approximately 10–20 days [9–11]. The biological factors that influence parasite developmental rates and ultimately the EIP are starting to become elucidated. The EIP has been shown to depend on environmental temperature and larval nutrition, with higher temperatures (up to a point) and plentiful nutrients available during larval stages accelerating parasite growth [8,12–16]. Our recent studies in *Anopheles gambiae*, one of the most effective malaria vectors in sub-Saharan Africa, have shown the *P. falciparum* EIP is also dependent on mosquito oogenesis, a process largely orchestrated by the steroid hormone 20-hydroxyecdysone (20E) [17]. When we reduced egg development using several means, including impairing 20E function and decreasing lipid mobilization, we detected the presence of larger oocysts in the midgut, as measured by averaging their cross-sectional area following mercurochrome staining. In turn, this led to infectious sporozoites reaching the salivary glands at earlier time points, leading to a shorter EIP [17]. Furthermore, faster oocyst growth in conditions of reduced egg development was characterized by the accumulation of neutral lipids in the midgut and was reversed by

depleting the major lipid transporter Lipophorin (Lp), suggesting that excess lipids that are not mobilized from the midgut mediate accelerated development. In agreement with this observation, a negative correlation between mean oocyst size and egg numbers indicated that parasites effectively exploit available mosquito resources for growth following egg development [17].

Given this unexpected relationship between oogenesis and the *P. falciparum* EIP, additional blood meals a female takes in order to complete further cycles of egg development may have important consequences for parasite transmission. Indeed, many mosquito species, including *An. gambiae*, naturally undergo multiple reproductive cycles of blood feeding, egg development and egg laying in their lifespan, which can be counted by ovarian dilatations [18,19]. Moreover, multiple feeds may be required even within a single reproductive cycle due to interrupted feeding or to nutrient deprivation during the larval stage (pre-gravid behavior) [20]. Additional blood meals may therefore potentially influence oocyst growth and the EIP, as suggested by reports showing an additional feed can increase *P. falciparum* oocyst size [21,22] and salivary gland sporozoite numbers at a given time point [22,23], and can be employed to boost sporozoite yields [24–26].

Here, we examine how parasite development is affected by multiple feedings, and illustrate the consequences of these effects using a simple epidemiological model of malaria transmission potential. By providing a second uninfected blood meal to females previously infected with *P. falciparum*, we show a striking increase in oocyst growth rates, which causes faster accumulation of sporozoites in the salivary glands and a substantially shortened EIP. When considered in the context of the basic reproduction number ($R_0$)—the average number of infections resulting from a first case—this shortened EIP leads to a consistent increase in malaria transmission potential in sub-Saharan Africa. Accelerated growth after an additional feeding event is not mediated by Lp-transported lipids, but is further enhanced in reproduction-defective females, suggesting that mosquito control strategies that reduce the reproductive output of *Anopheles* females—such as population suppression gene drives—could actually favor parasite transmission. These data have important implications for accurately understanding malaria transmission potential and estimating the true impact of current and future mosquito control measures.

## Results

### An additional blood meal accelerates *P. falciparum* oocyst development in an Lp-independent manner

We set out to determine the potential effects of a second blood feeding on parasite development, and whether any of these effects might be mediated by the lipid transporter Lp. To achieve this, we injected *An. gambiae* females with double-stranded RNA (ds*RNA*) targeting *Lp* (ds*Lp*) and compared them to control females injected with dsRNA targeting *green fluorescent protein* (ds*GFP*) (**S1 Fig**). After allowing females to mate, we blood fed them on a *P. falciparum* (NF54) culture and provided them with the opportunity to lay eggs, and then, at 3 days (d) post infectious blood meal (pIBM), we gave them an additional, uninfected blood feed (2BF groups, **Fig 1A**). By this time, females have completed their first gonotrophic cycle and are therefore ready to feed again to produce a second egg batch. For comparison, feeding control groups were instead maintained on sugar after the initial infectious feed (1BF groups, **Fig 1A**). We dissected females from all four groups (ds*GFP* 1BF; ds*GFP* 2BF; ds*Lp* 1BF; ds*Lp* 2BF) at 7 d pIBM, and analyzed oocyst numbers and size by fitting linear mixed models incorporating the number of blood feeds, ds*RNA* injection and their interaction as fixed effects, and replicate as a random effect (**S1 Table**). While we detected no effects on the prevalence ($\chi^2$ test, **S3**

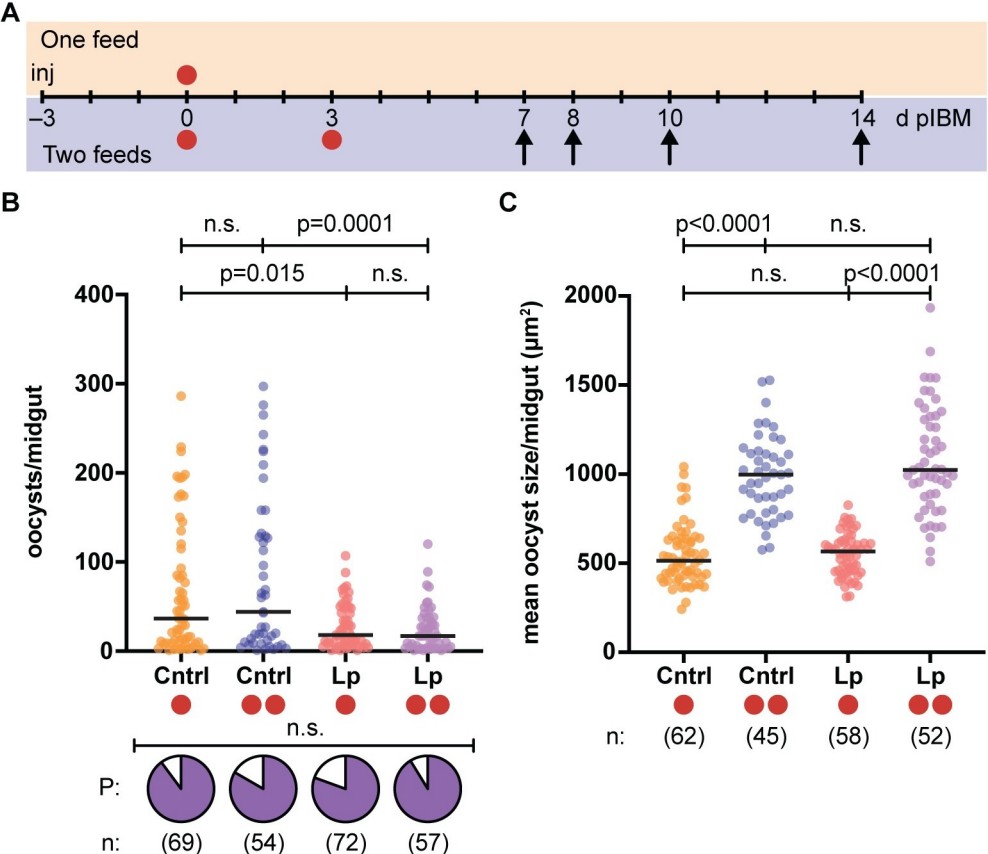

**Fig 1. A second blood meal increases oocyst size in a Lp-independent manner.** (A) ds*GFP* (Cntrl)- and ds*Lp*-injected females were infected with *P. falciparum* 3 days post injection (inj) and then either provided a second uninfected blood meal 3 days post infectious blood meal (d pIBM) (two red circles) or maintained on sugar (one red circle). Infection outcomes in all groups were determined at 7–14 d pIBM (arrows). (B) Oocyst prevalence (P, pie charts) ($\chi^2$ test: $\chi^2$ = 4.3, d.f. = 3, n.s.) and intensity (Linear mixed model; FDR-corrected post-hoc Student's t tests shown for all models) are not affected by a second blood meal at 3 d pIBM (#BF: n.s.), but intensity is lower in Lp-depleted groups (ds*RNA*: p<0.0001). (C) Oocyst size at 7 d pIBM is significantly increased in females fed twice in both injection groups (Linear mixed model; #BF: p<0.0001; ds*RNA*: n.s.). Horizontal bars indicate median values. n = numbers of mosquitoes analyzed from 3 different experiments. n.s. = not statistically significant. See S1–S4 Tables for details of statistical models.

Table) or intensity of infection (**Fig 1B**, prevalence shown in pie charts), we observed a striking increase in oocyst area in females that had been blood fed a second time, measured by averaging the cross-sectional area of up to 50 mercurochrome-stained oocysts in each midgut (**Fig 1C**). Median oocyst size was 94% larger in these females (model: #BF: p<0.0001), suggesting increased parasite growth rates following an additional blood meal. However, increased oocyst size was not mediated by Lp-transported lipids, as after a second blood meal oocysts in *Lp*-silenced females showed an increase in size comparable to those in ds*GFP* females (model: ds*RNA*: n.s.; **Fig 1C**). *Lp* silencing did reduce oocyst numbers, but without affecting prevalence, as previously observed after a single blood meal (model: ds*RNA*: p<0.0001; **Fig 1B**) [17,27].

We next performed immunofluorescence microscopy to assess the developmental stage of these oocysts using a DNA stain and an antibody recognizing the *P. falciparum* circumsporozoite surface protein (CSP), which is expressed during late oocyst development and on sporozoites [28]. At 8 d pIBM, while oocysts developing in females fed once had DNA but no

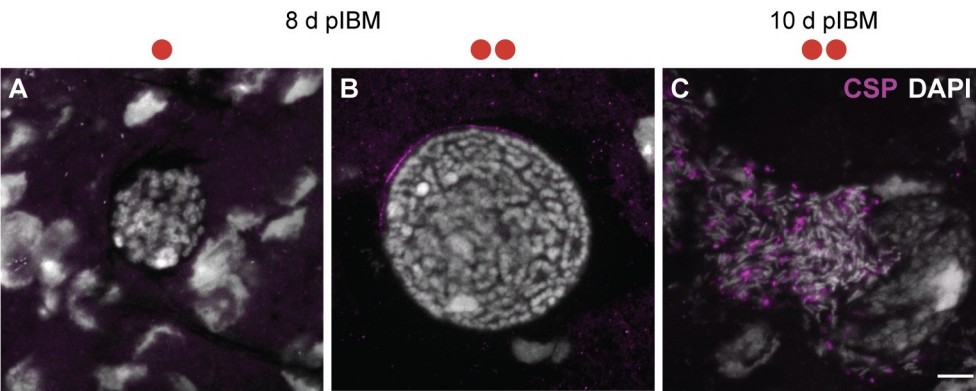

**Fig 2. A second blood meal accelerates oocyst development.** (A–B) Immunofluorescence assay of oocysts from control females (A) fed once (one red circle) or (B) fed twice (two red circles) at 8 d pIBM. (C) Oocyst from a female fed twice showing the release of mature sporozoites at 10 d pIBM. Sporozoites are labelled with circumsporozoite protein CSP (magenta) and DNA is stained with DAPI (gray). In all panels, scale bar = 10 μm.

detectable CSP staining, oocysts derived from females that had fed twice already showed increased DNA content and detectable CSP expression (**Fig 2A and 2B**), consistent with additional rounds of DNA replication and the beginning of sporozoite formation. Remarkably, by 10 d pIBM, some oocysts in this group were already releasing mature CSP-labelled sporozoites (**Fig 2C**). Together these data show that *P. falciparum* development is accelerated when females blood feed a second time, and that accelerated growth is not significantly mediated by the lipid transporter Lp.

## Mosquitoes are infectious sooner following an additional blood meal

Accelerated parasite development suggests that mosquitoes may become infectious sooner following an additional blood meal, with consequences for their transmission potential. We therefore compared the timing of the appearance of sporozoites, the infectious stage of parasite development, in dissected salivary glands over the course of several days of sporogony in ds*GFP*-injected females fed once or twice. Following a similar protocol of a second blood meal at 3 d pIBM, we detected sporozoites as early as 7 d pIBM, and we observed a significant increase in sporozoite prevalence at 8 d pIBM—an early time for sporozoite invasion of the salivary glands—when we detected sporozoites in 33% of control 2BF females compared to 3% in control 1BFs (**Fig 3A**). This significant increase was still evident at 10 d pIBM, with 84% of ds*GFP* 2BF mosquitoes harboring sporozoites compared to 40% of 1BF females (**Fig 3A**). We calculated that females were 15-fold and 7.7-fold, respectively, more likely to have sporozoites in their salivary glands at these early time points if they had had an additional blood meal ($\chi^2$ test; 8d: p = 0.0018; 10d: p<0.0001; **S3 and S4 Tables**). Moreover, the intensity of infection was also increased at 10 d pIBM (model: #BF: p = 0.0001; **Fig 3B, S1 and S2 Tables**), where we observed a number of highly infected mosquitoes (>10,000 sporozoites/salivary glands), but low prevalence in the singly fed groups prevented statistical testing at 8 d pIBM (**S2 Fig**). These observations were not due to a change in the prevalence or intensity of oocyst infection, which were both unaffected in ds*GFP*-injected controls as described above (**Fig 1B**). By 14 d pIBM 1BF and 2BF mosquitoes became comparably infected, as sporozoites in 1BF mosquitoes had also reached the salivary glands, with sporozoite prevalence near 100% (**Fig 3A**) and only a significant increase in infection intensity across the 2BF groups (model: #BF: p = 0.034) that did not persist after post-hoc testing (**S2 Fig, S1 and S2 Tables**).

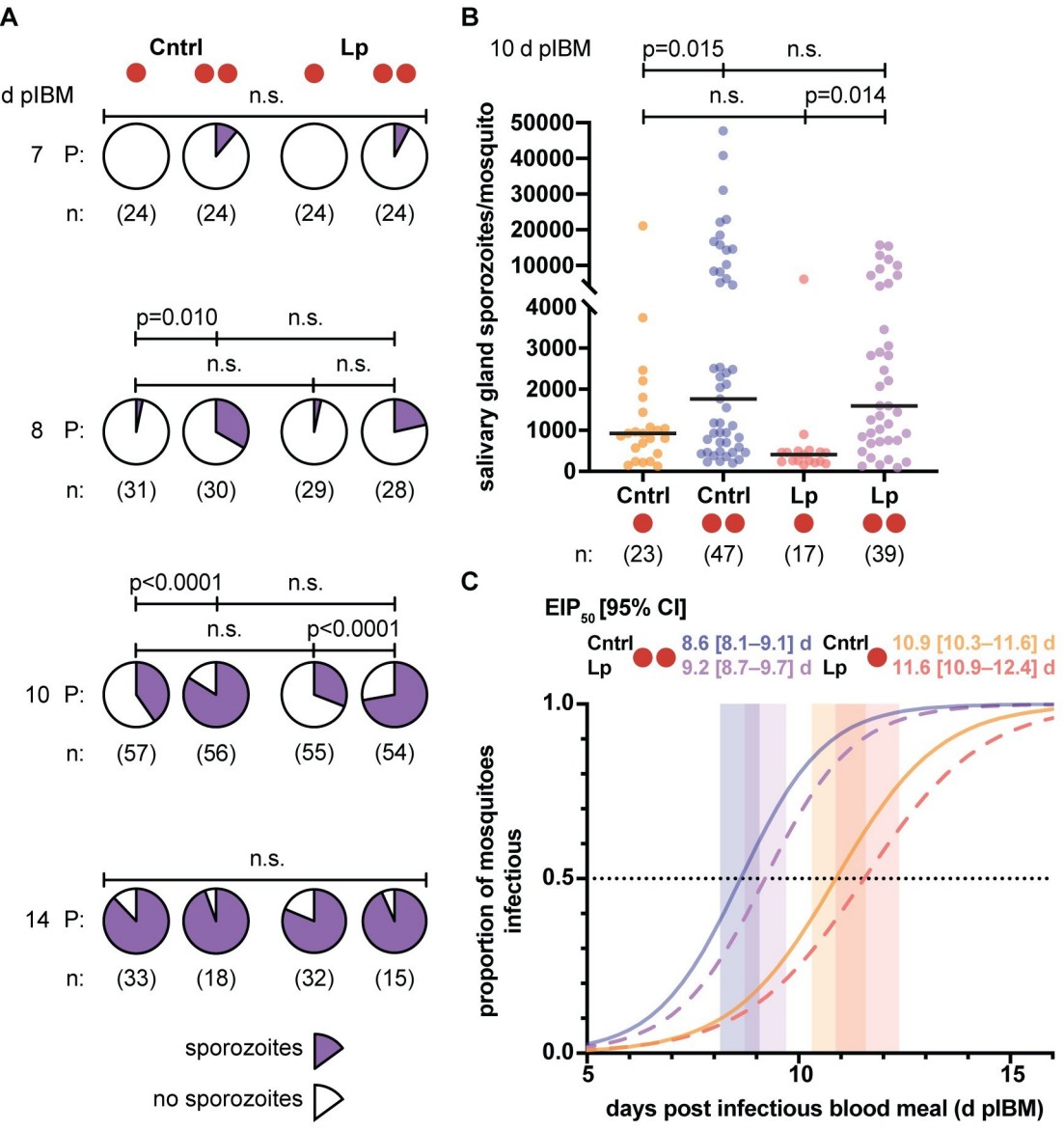

**Fig 3. Mosquitoes are infectious sooner following a second blood meal.** (A–B) Salivary glands of females fed twice (two red circles) show (A) a significantly higher prevalence (P, pie charts) of sporozoites at 8 and 10 d pIBM ($\chi^2$ test: 8 d, $\chi^2$ = 15, d.f. = 3, p = 0.0018; 10 d, $\chi^2$ = 43, d.f. = 3, p<0.0001; FDR-corrected post-hoc $\chi^2$ tests shown) and (B) significantly more sporozoites at 10 d pIBM (Linear mixed model; #BF: p = 0.0001; FDR-corrected post-hoc Student's t tests shown) than females fed once (one red circle). Horizontal bars indicate median values. There is no difference in sporozoite prevalence at a later time point ($\chi^2$ test: 14 d, $\chi^2$ = 2.4, d.f. = 3, n.s.). (C) The $EIP_{50}$ (time to 50% infectious–dotted line; values also shown ± 95% C.I.) of control (Cntrl) females fed a second time (blue solid line) is reduced by 2.3 d (21%) compared with controls fed once (orange solid line), as determined from the sporozoite prevalence data shown in (A) (z test, Z = 5.7, p<0.0001). Similarly, $EIP_{50}$ is reduced by 2.4 d (20%) in Lp-depleted females fed a second time (purple dashed line) compared to those fed once (red dashed line) (z test, Z = 5.3, p<0.0001). Fitted logistic curves (lines), $EIP_{50}$ ± 95% C.I. (shaded). n = numbers of mosquitoes analyzed from 4 different experiments. n.s. = not statistically significant. See S1–S4 Tables for details of statistical models.

Consistent with the observed lack of effects on oocyst growth (**Fig 1C**), *Lp*-silenced females behaved as the ds*GFP* controls and also showed a striking increase in prevalence of infection after a second blood meal (**Fig 3A, 3B** and **S2 Fig**). Although the reduced oocyst numbers lowered total sporozoite intensities at 14 d pIBM (model: ds*RNA*: p = 0.0053), again this difference was not significant after post-hoc testing (**S2 Fig**).

To compare the rates of parasite development between our treatment groups, we constructed EIP curves based on the above prevalence data using logistic regression to describe over time how an additional blood meal changes the infectiousness of mosquitoes (**S3 Table**). From these curves, we calculated the $EIP_{50}$—the point at which half of the mosquitoes become infectious, or in other words the median time until the appearance of sporozoites in the salivary glands—for each treatment group. An additional blood feeding significantly shortened the $EIP_{50}$, with ds*GFP* 2BF mosquitoes becoming infectious 2.3 ± 0.4 d earlier than ds*GFP* 1BF (Z test: p<0.0001; **S4 Table**), corresponding to a 21% reduction in EIP (**Fig 3C**, solid lines). Larger oocyst size at 7 d pIBM is therefore associated with shorter estimates for the EIP, as previously shown [17]. Similar results were obtained in Lp-depleted females (2.4 ± 0.4 d; 20% reduction; p<0.0001), confirming that neutral lipids carried by this transporter do not mediate increased parasite growth after a second blood feeding event (**Fig 3C**, dashed lines). Given mosquitoes regularly take multiple (>3) blood meals in the field [9,29], these results suggest that transmission is likely to occur much sooner than previously thought, and is therefore also mediated by younger mosquitoes.

## Modelled estimates of $R_0$ with a single blood meal underestimate transmission

To illustrate the relevance of these results for malaria transmission, we chose a simple metric of transmission potential—the basic reproductive number $R_0$, calculated as a function of temperature [30]. We mapped the consequence of a second blood meal on the distribution of $R_0$ across sub-Saharan Africa, considering all locations where *An. gambiae* and closely related *Anopheles* vectors are present. We chose to use this particular temperature-dependent model formulation because $R_0$ depends exponentially on the EIP and the mosquito death rate, which are both influenced by temperature. To avoid over-extrapolating our laboratory data, we estimated $R_0$ in locations within a narrow temperature range (27 ± 2˚C) around our experimental conditions (see Materials and Methods).

For each 5x5 km grid cell of the map, we calculated both the standard $R_0$ using $EIP_{50}$ estimates derived from a single blood feed and the adjusted $R_0$ ($R_0^b$) using the proportional reduction in $EIP_{50}$ observed in our ds*GFP* 2BF females (21%). We then mapped the ratio of the adjusted to standard $R_0$ in each grid cell across sub-Saharan Africa for each month of the year within our temperature range. The number of months that mean monthly temperature falls within this range are presented (**Fig 4A**). The area within this temperature range for at least one month of the year is home to nearly 738 million people, roughly half of the population in Africa.

The reduction in $EIP_{50}$ leads to higher modeled mean $R_0$ in all mapped locations. The average increase in $R_0$ is 10.5% (range: 10.1%–12.1%) across all regions of sub-Saharan Africa with at least one month with a mean temperature within 27 ± 2˚C (**Fig 4B**). This result implies that epidemiological models directly incorporating currently accepted EIP parameters may be systematically underestimating malaria transmission potential across a substantial fraction of sub-Saharan Africa.

## Effects of an additional blood meal are exacerbated in females with impaired egg development

We have previously shown that in females blood fed once, parasite development is substantially faster in multiple instances where oogenesis is reduced [17]. As mosquito reproduction is a target of genetic control strategies currently in the pipeline [31,32], we went on to determine whether parasite growth could be further boosted after a second blood meal in females with

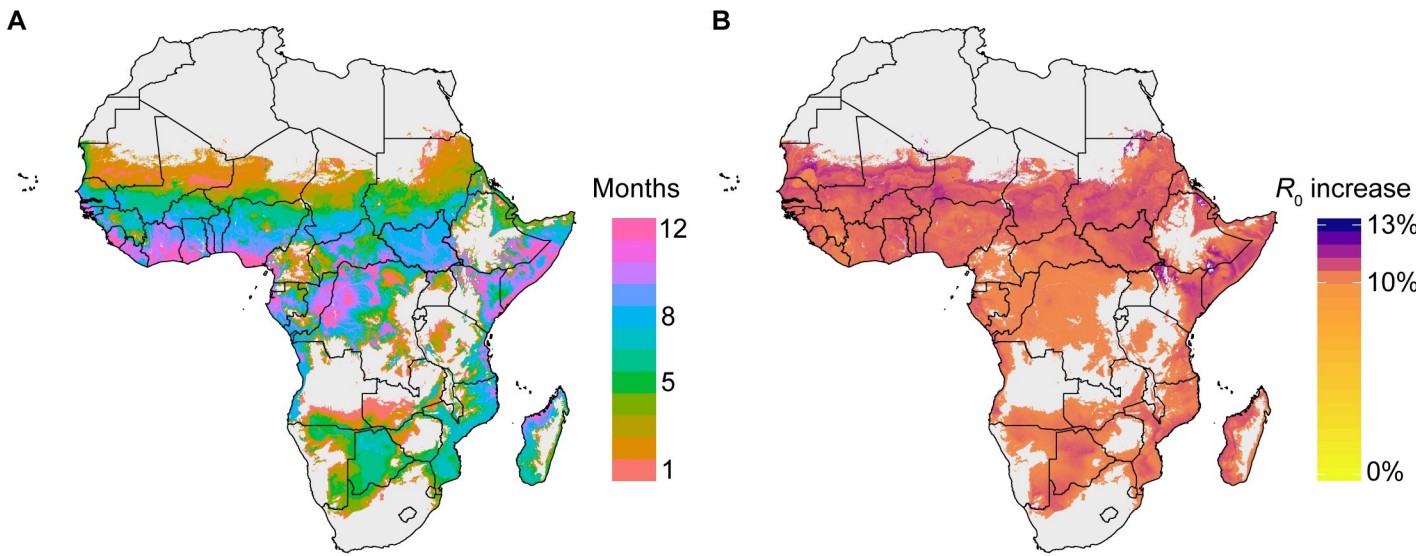

**Fig 4. Relative $R_0$ values across sub-Saharan Africa show increase in malaria transmission potential.** (A) Regions of sub-Saharan Africa with 1 to 12 months of average temperature in 27 ± 2°C. Our modeling analysis was restricted to locations and months of the year during which the average temperature was in this range, in order to align with our laboratory conditions. Areas in gray depict regions where there are 0 months with average temperature in that range, or where the predicted probability of *An. gambiae* complex is less than 5%. (B) The percent increase of $R_0^b$ using revised estimates of $EIP_{50}$ (derived from two blood feeds) to $R_0$ using the standard estimates of $EIP_{50}$ (derived from a single blood feed). Models using a standard EIP parameter may underestimate $R_0$ by an average of least 10.1%. In these regions, the average change during the months within the relevant temperature range is 10.5% (range: 10.1%–12.1%).

impaired egg development. To this aim, we used genetically modified *An. gambiae* mutants that are mosaics of *zero population growth* (Δ*zpg*), a gene required for germline cell maintenance whose disruption in females causes severely atrophied ovaries [17,33]. After infecting Δ*zpg* or control females with *P. falciparum*, we replicated the experimental design described above and provided an additional blood meal to approximately half of each group to compare parasite development in both feeding regimes. Median oocyst size was increased in Δ*zpg* females fed either once or twice when compared to controls (model: #BF: p<0.0001, genotype: p<0.0001; **Fig 5A**, **S1** and **S2** **Tables**), confirming that *P. falciparum* growth can be further boosted in this background, while oocyst prevalence was unaffected (**Fig 5B**, pie charts). Despite lower oocyst numbers (genotype: p<0.0001; **Fig 5B**), sporozoite intensity in the salivary glands at an early time point (10 d pIBM) was higher in Δ*zpg* females than in controls under both feeding regimes, and significantly so after one blood feed (model: #BF: p<0.0001, genotype: p = 0.0056; **Fig 5C**). Moreover, sporozoite prevalence at 10 d pIBM was significantly higher in Δ*zpg* females fed once ($\chi^2$ test; p<0.0001; **Fig 5C**, upper pie charts). Finally, cumulative prevalence across both single and double blood feedings was significantly increased (Fisher's exact test: p<0.0001) and these mutants were 3.9-fold more likely to have sporozoites in their salivary glands than control mosquitoes (odds ratio; 95% CI: 2.3–6.5 fold; **Fig 5C**, lower pie charts). These results suggest that, even in cases where they may support lower oocyst loads, females with impaired egg development might be more effective at transmitting *P. falciparum* parasites under either blood feeding regime.

## Discussion

The parasite EIP is a key parameter in malaria transmission dynamics. Given the relatively short *Anopheles* lifespan—estimated to be around 10–20 days depending on species and environmental conditions [9–11]—parasites with faster sporogonic development are more likely to

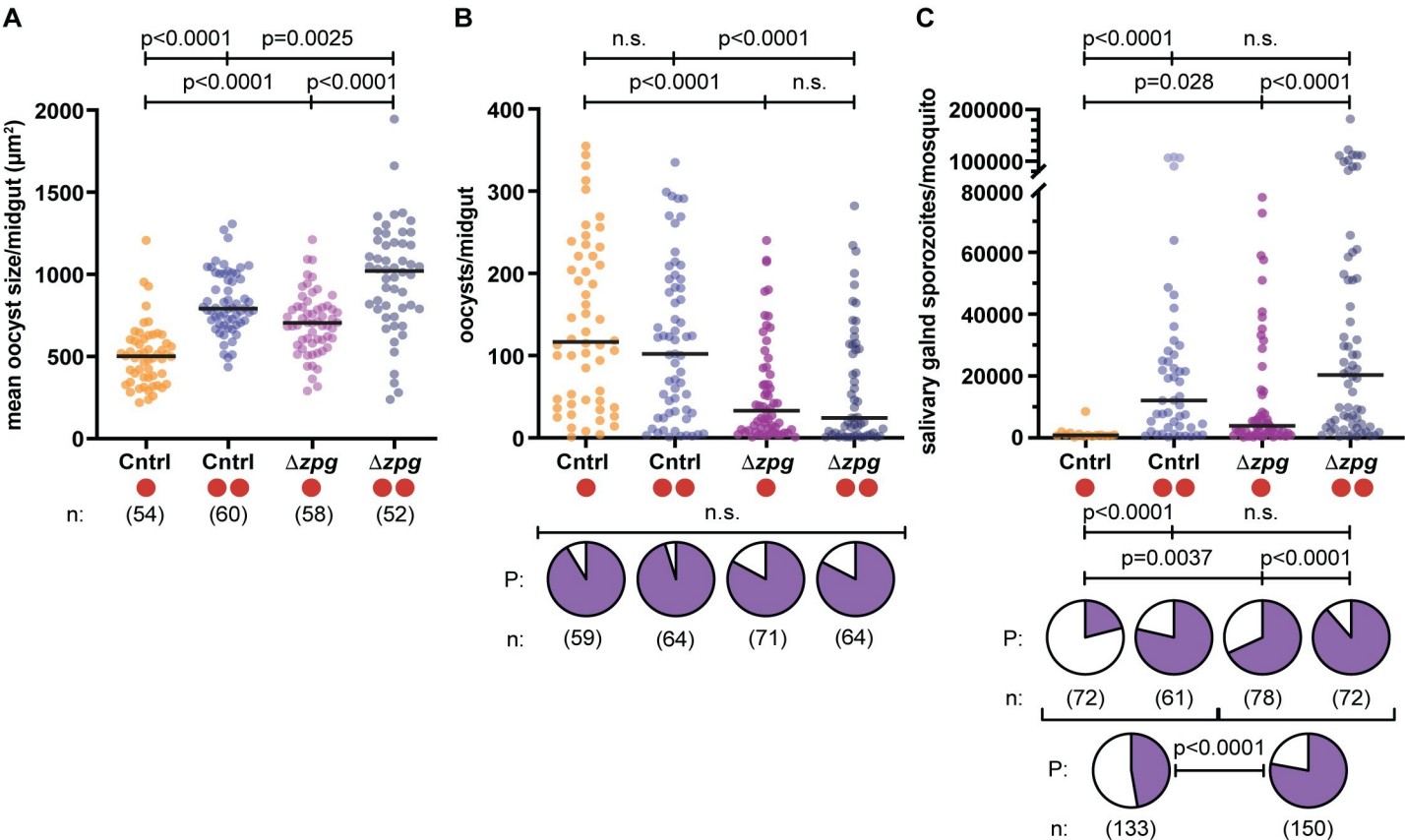

**Fig 5. Parasite developmental rates are further enhanced in eggless mosquitoes.** (A) Oocysts are significantly larger in Δ*zpg* (eggless) mutant females at 7 d pIBM after both one or two blood meals (Linear mixed model; #BF: p = 0.0001; genotype: p<0.0001; FDR-corrected post-hoc Student's t tests shown for all models) compared to controls (Cntrl). (B) Oocyst intensities are lower in Δ*zpg* mutant females compared to controls (Linear mixed model; #BF: p = 0.041; genotype: p<0.0001), whereas oocyst prevalence (P, pie charts; χ$^2$ test: χ$^2$ = 6.5, d.f. = 3, n.s.) is unaffected. (C) Salivary glands sporozoite numbers are increased in both control and Δ*zpg* mutant females at 10 d pIBM after a second blood meal (Linear mixed model; #BF: p<0.0001; genotype: p = 0.0056), with sporozoite prevalence (P, upper pie charts) also increased in the Δ*zpg* mutant background (χ$^2$ test, χ$^2$ = 81, d.f. = 3, p<0.0001), significantly at the first blood meal. Pooling data shows higher sporozoite prevalence in Δ*zpg* mutant population (P, lower pie charts) (Fisher's exact test). Horizontal bars indicate median values. n = numbers of mosquitoes analyzed from 3 (A and B) or 4 (C) different experiments. n.s. = not statistically significant. See S1–S4 Tables for details of statistical models.

be transmitted to the next human host [4]. Here, we demonstrate that multiple blood feedings significantly accelerate parasite growth, shortening the time required for sporozoites to appear in the salivary glands. These findings are consistent with infection studies in closely related mosquito species that showed that an additional blood meal during development either boosts *P. falciparum* sporozoite intensities or yields [22–26,34] or increases oocyst size [21–23], although none analyzed parasite growth rates after the second meal. Our study conclusively links the effects of an additional blood meal to increased parasite developmental rates and a shortened EIP. Future work should determine whether additional feeding events during the same gonotrophic cycle also impact *P. falciparum* growth, or whether the presence of parasites in a second blood meal would affect the development of already established parasites.

How does this accelerated development occur? An additional blood meal provides the female mosquito with greater nutrient resources, especially amino acids and lipids, that both could be potentially transported across the oocyst membrane. Parasites require lipids to generate phospholipid-containing membranes for subdivision into thousands of sporozoites, rather than for energy (as they seem to lack canonical enzymes for β-oxidation [35]). Oocysts can

take up labelled mosquito lipoparticles [36], and increased availability of lipids as provided by an additional blood meal may accelerate host lipid scavenging, which also occurs during human stages of infection [37–39]. In this study, however, depletion of Lp—the major mosquito lipid transporter—did not slow rates of oocyst growth or sporozoite appearance after a second blood feeding, with only a slight, non-significant delay in $EIP_{50}$ detected, suggesting that other nutrients (possibly amino acids, as suggested by a recent report [21]) may mediate the bulk of the observed boost in oocyst development. *Lp* silencing also had no effects on *P. falciparum* oocyst growth in females fed once (**Fig 1C**) [17], contrary to observations in the mouse malaria model *P. berghei* [40,41]. Interestingly, however, in our previous studies we showed that Lp-transported lipids do mediate the accelerated growth observed in conditions of reduced egg numbers, likely linked to the observed accumulation of lipids in the midgut in those females [17]. Combined, these results point to a remarkable plasticity in *P. falciparum* parasites in utilizing mosquito nutrients for growth depending on the metabolic state of the *Anopheles* female, and suggest fundamental differences in the mechanisms mediating development of human versus rodent malaria parasites. In future studies it will be interesting to ascertain the mechanisms behind the reduced oocyst numbers observed in Lp-depleted females, and if those are similar to the mechanisms induced in Δ*zpg* mutants.

Consistent with accelerated oocyst growth and maturation, we observed an earlier presence of sporozoites in the salivary glands and a significant decrease in the EIP. Considering females blood feed on average every 2–3 days [9], and thus have taken >2 meals by the time they become infectious [29], our results may represent a conservative estimate of the impact of feeding behavior on the EIP. Indeed, our data using Δ*zpg* mutants demonstrate that *P. falciparum* growth rates are not saturated with two blood feeds and that parasites are capable of undergoing even shorter developmental cycles in the mosquito vector. Although we did not see a significant increase in sporozoite intensities in Δ*zpg* females fed twice compared to controls under the same feeding regime, we observed a number of females at 10 d pIBM with very high infection intensities (>10,000 sporozoites/salivary glands) (**Fig 5C**). Based on a recent report demonstrating a similar intensity threshold is required for infectivity of *P. yoelii* sporozoites to the mammalian host [42], our data suggest the earlier appearance of more highly infected mosquitoes may increase the force of transmission, especially if a few highly infected mosquitoes are the primary drivers of transmission. However, the overall impact of additional blood meals in malaria endemic areas will likely also depend on other environmental factors affecting lifespan and blood feeding frequency.

One caveat of our study is that we measured the effects of an additional blood meal only in *An. gambiae* and we cannot exclude that the magnitude of the decrease in EIP will be different in other *Anopheles–Plasmodium* combinations. Additional blood meals, however, have been shown to shorten the dengue virus EIP [43], improve *Leishmania* transmission [44], accelerate filarial worm development [45] and increase La Crosse virus replication [46]. All these findings suggest that exploiting mosquito resources may be a generalized mechanism for pathogens to plastically accelerate their development and improve their odds of transmission.

While transmission efficiency may be assessed directly from epidemiological data surveys [47], estimations of potential interventions require models informed by entomological data. Incorporating our shortened EIP into a simple model of malaria $R_0$ to reflect multiple blood feeding, we see the $R_0$ increases by an average of 10.5% across much of sub-Saharan Africa. The neglected contribution of multiple blood feeding may partly explain higher malaria burdens in places and seasons when estimated transmission potential is low [48]. Moreover, underestimates of malaria transmission potential would suggest that malaria elimination efforts in many settings may be more challenging than anticipated. One limitation is our choice of the basic reproductive number, which considers mean-field dynamics, without

accounting for known heterogeneities such as differences in mosquito lifespan, parasite strain competition and spatial variation [49]. While these heterogeneities play a role in determining the precise level of transmission, we expect them to affect our transmission estimates equivalently and in the same direction under both single and multiple blood feeds. Thus, in our calculation of the ratio of $R_0$s to estimate the impact of a second blood feeding, the effect of these heterogeneities in a given location will cancel out. Importantly, although many estimates of $R_0$ are derived from prevalence and thus may not directly incorporate EIP, others do explicitly use this parameter and thus may be systematically underestimating its relevance.

What are the implications of these findings for vector control strategies? Firstly, our data point towards females potentially contributing to malaria transmission from a younger age. Given the observed age-dependent mortality induced by insecticides, with higher resistance levels observed in younger mosquitoes [50,51], this observation implies that current models of insecticide-based interventions (both LLINs and IRS) need to be revisited. When combined with the observation that multiple blood feeding increases insecticide resistance in older females of a related species, *An. arabiensis* [52], our results suggest that mosquitoes feeding multiple times are more likely to survive to the point when they become infectious. Previous work has similarly predicted a lesser effectiveness for mosquito control methods targeting survival when the EIP is shorter [53].

Secondly, our results obtained with Δ*zpg* mutants show that—due to a further increase in parasite growth rates—mosquitoes with reduced reproductive capacity have a significantly higher transmission potential, even in the face of possible lower oocyst numbers. These findings are particularly relevant for control strategies that tamper with mosquito reproduction, such as genetically-engineered population suppression gene drives [31,32]. At a time when *Anopheles* gene drive strains are being tested in semi-field settings, our data call for a careful evaluation of whether these genetically-modified mosquitoes would contribute to more efficient malaria transmission while the drive is spreading.

Finally, our results emphasize that informed policy decisions on current and future malaria control strategies can only be built on thorough research into the fundamental factors affecting malaria transmission biology.

## Materials and methods

### *Rearing of* Anopheles gambiae *mosquitoes*

*Anopheles gambiae* mosquitoes (wild-type G3 and transgenic strains) were reared in cages at 27˚C, 70–80% humidity on a 12 h light:12 h dark cycle. Adults in colony cages were fed on 10% glucose solution *ad libitum* and weekly on human blood (Research Blood Components, Boston, MA). Males and females were sexed as pupae and mated in large cages.

*Plasmodium falciparum* (NF54 strain) was cultured as in [17] and is used under the permissions of a material transfer agreement from the laboratory of Carolina Barillas-Mury, National Institutes of Health, Bethesda, MD, USA.

### *Gene expression knockdown using ds*RNA

PCR fragments of the eGFP control (495 bp) and Lp (600 bp) were amplified from plasmids pCR2.1-eGFP and pLL10-Lp, as described previously [17,54] and verified by DNA gel electrophoresis. ds*RNA* was transcribed and purified from the PCR templates using the Megascript T7 transcription kit (Thermo Fisher Scientific) as described previously [55]. 690 ng of ds*RNA* (ds*GFP*, ds*Lp*) was injected (Nanoject II, Drummond) at a concentration of 10 ng/nl into adult G3 females within 1 day (d) of eclosion. Females were randomly assigned to injection groups and surviving females were mated with G3 males and used in subsequent experiments. Gene

knockdown levels were determined in at least 3 biological replicates by RNA extraction, cDNA synthesis and quantitative real-time PCR at 6 d post injection as described in [17].

### P. falciparum *infections of* An. gambiae *mosquitoes*

Cages of mated female mosquitoes aged 4 d (or 4–6 d for transgenic mosquitoes) were blood fed on ~320 µl *P. falciparum* culture for 30–60 min via heated membrane feeders and introduced into a custom-built glove box (Inert Technology, Amesbury, MA). Feeding behavior was encouraged by starving mosquitoes of 10% glucose solution for 24 h and females not fully engorged were removed. Blood-fed mosquitoes were provided 10% glucose solution for 48 h and then given an oviposition site. Females were fed a second time 3 d after the initial infectious blood meal, using uninfected blood. Blood intake at the second blood meal was encouraged by providing an oviposition site 2 d post infectious blood meal (pIBM) and non-blood fed mosquitoes were removed. Females blood-fed twice and control mosquitoes fed once were provided 10% glucose solution until dissection. At dissection time points, mosquitoes were aspirated into 80% ethanol and transferred to 1X phosphate-buffered saline (PBS) (oocyst stages) or aspirated into ice-cold PBS (sporozoites stages) and beheaded. At least 3 biological replicates of each infection were performed.

Oocyst counts and measurements: At 7 d pIBM, midguts were stained directly in 2 mM mercurochrome (Sigma-Aldrich, St. Louis, MO) for 12 min. Mercurochrome-stained midguts were imaged at 100X on an Olympus Inverted CKX41 microscope, and oocysts were counted and measured using scaled images in FIJI [56]. Burst oocysts were counted but excluded from oocyst size analysis. Mean oocyst size was calculated for each midgut to avoid pseudoreplication.

Oocyst staining: At 8 and 10 d pIBM, midguts were fixed in 4% formaldehyde for 30–40 min, permeabilized and blocked for 1 h in PBS, 0.1% Triton (PBS-T), 1% bovine serum albumin (BSA) at 22˚C, and stained with an anti-CSP mouse monoclonal 2A10 (BEI Resources) (1:350), followed by a goat anti-mouse-Alexa 488 secondary antibody (Molecular Probes) (1:1000). Samples were washed in PBS-T, stained with DAPI (1 µg/ml) and mounted in Vectashield with DAPI (Vector Laboratories, Burlingame, CA), and then imaged at 630X on a Zeiss Inverted Observer Z1 with Apotome2. Scaled images were processed in FIJI.

Sporozoite counts: Mosquitoes were decapitated and the salivary glands of individual females were collected in a small volume of PBS and disrupted using a handheld disposable pestle. Released sporozoites were spun at 8000 *g* for 10 min at 4˚C and resuspended in a known volume of PBS. Sporozoites in 0.1 µl were counted using a disposable hemocytometer at 200X magnification on an Olympus Inverted CKX41 microscope with phase-contrast microscopy and sporozoite totals for each mosquito were calculated.

### *Generation of Δzpg females*

Zpg and Cas9-carrying mosquito strains were generated previously [17]. Zpg/Cas9 mutant females, hereafter Δ*zpg* mutants, were the F1 progeny of Zpg and Cas9 transgene homozygotes. Zpg controls and Δ*zpg* mutant females were mated to G3 males prior to infection, and did not differ in their ability to take a blood meal.

### Statistical analysis

Experimental data were analyzed using JMP 14 Pro statistical software and GraphPad Prism 8.0. JMP 14 Pro was used to construct statistical models to account for variation due to multiple factors in an experiment. Residual Maximum Likelihood (REML) variance components analysis was used by fitting linear mixed models after data transformation to normality. The

number of blood feeds, ds*RNA* injection or genotype and their interaction were used as fixed effects and replicate was included as a random effect. Model effect test outputs are reported and multiple comparisons were calculated using 4 pairwise Student's t tests followed by FDR correction (**S1 and S2 Tables**). Graphpad Prism 8.0 was used to calculate EIP (using logistic regression curves), Z tests, Fisher's exact tests and $\chi^2$ tests (**S3 and S4 Tables**). A significance value of 0.05 was used as a threshold in all tests.

## Modeling

We used a previously published model [30] of the basic reproductive number, $R_0$, incorporating temperature dependence, to explore the implications of shortening EIP alone on estimated transmission potential. With this model we aimed not to predict $R_0$ across regions considered, but rather to examine how the EIP could impact $R_0$ estimates. To minimize the extrapolation of our laboratory experiments, we estimate $R_0$ in locations within a narrow temperature range (27 ± 2˚C) around our experimental conditions.

To demonstrate the impact of a shorter EIP on $R_0$, we use equation (2) from [30] simplified here as

$$R_0(T) = \sqrt{f(T)e^{-g(T)}},$$

where $f(T)$ and $g(T)$ are functions of temperature-dependent trait data from *Anopheles* species and $g(T)$ depends linearly on EIP (**S1 Text**, **S3 Fig**, **S5** and **S6** Tables). We then modified the basic reproductive number to incorporate a second blood feeding by scaling EIP using the term $\beta$, which we refer to as $R_0^b$ given by

$$R_0^b(T) = \sqrt{f(T)e^{-\beta g(T)}}.$$

We determine the scaling parameter $\beta$ as the relative reduction in EIP in the presence of a second blood feed: $\beta$ = 8.63 (2BF)/10.88 (1BF) = 0.793. The change in $R_0$ using a shortened EIP is shown by the ratio of the modified $R_0$ to the original $R_0$ as

$$\frac{R_0^b}{R_0}$$

(**S4 Fig**). To estimate the impact of the EIP reduction on transmission potential, we applied these functions to current human population and monthly mean temperature data across sub-Saharan Africa to estimate the percent change in $R_0$ under a scenario where mosquitoes blood feed while infected. As our laboratory data were collected at 27˚C and many of the drivers of malaria transmission are sensitive to temperature, we restricted our $R_0$ projections to the regions and months of the year where the average temperature was 27 ± 2˚C (**Fig 4A**).

We used spatial raster data to calculate the change in $R_0$ across relevant regions of sub-Saharan Africa. We obtained global monthly minimum and maximum temperatures for the period 2010–2018 from TerraClimate [57]. Similar to Fick *et al.* [58], we calculated the average temperature for each month during this period, by taking the mean of the recorded maximum and minimum temperature. We then determined the average temperature for each calendar month (Jan–Dec) by taking the mean for that month over this nine-year period. We restricted our modeling analysis only to months and locations with mean temperatures between 27 ± 2˚C. We further restricted analysis to the regions of Africa where the predicted probability of *An. gambiae* and closely related species (*An. gambiae* complex) is greater than 5%, using predicted *Anopheles* distribution maps from the Malaria Atlas Project [59], under the assumption that the observed effects on the EIP are consistent in these species. Spatial human

population distribution for 2020 was obtained from WorldPop [60]. Monthly mean temperature and human population data were aggregated with bilinear resampling to match the projection of the *Anopheles* geographic extent data at a 5x5 km resolution.

Using these data, we calculated $R_0$ and $R_0^b$ for each 5x5 km grid cell for each month. We determined the ratio $R_0^b/R_0$ for each month to show the monthly change in transmission potential under a multiple blood feeding scenario (**S5 Fig**). We considered the mean of the monthly $R_0$ and $R_0^b$ estimates during months at 27 ± 2˚C and examined the ratio of the means. To evaluate the relevance of our results to the population of sub-Saharan Africa, we tallied the total population living in the mapped areas, that is, 5x5 km grid cells with at least one month in the 27 ± 2˚C temperature range, and find nearly 738 million people living in these regions.

The modelling code is available on Github at https://github.com/iholmdahl/DoubleBloodFeeds.

## Supporting information

**S1 Text. Modeling details.**
(PDF)

**S1 Fig. *Lp* expression is effectively silenced following ds*RNA* injection.** *Lp* gene expression was determined in pools of 5–10 decapitated females at 6 d post injection (3 d pIBM) at the time of the second blood feed. *Lp* expression levels were normalized to *Rpl19*. Four biological replicates were analyzed with means ± standard error shown by horizontal bars. Unpaired t-test.
(TIF)

**S2 Fig. Sporozoite intensities in control and Lp-depleted mosquitoes at 8 and 14 d pIBM.** (A) Salivary glands of females fed twice (two red circles) show more sporozoites than females fed once (one red circle) at 8 d but low prevalence in singly-fed groups prevents a determination of statistical significance. (B) Sporozoite levels in salivary glands at 14 d pIBM are comparable between singly and doubly-fed control and *Lp*-silenced mosquitoes (Linear mixed model; #BF: p = 0.034; ds*RNA*: p = 0.0053; FDR-corrected post-hoc Student's t tests shown). Neither the increase in infection intensity across the 2BF groups, nor the decreased sporozoite intensity in *Lp*-silenced groups persist after post-hoc testing (S1 and S2 Tables). Horizontal bars indicate median values. n = numbers of mosquitoes analyzed from 3 different experiments. n.s. = not statistically significant.
(TIF)

**S3 Fig. Anopheles gambiae EFD data fit to a quadratic function.** Data points extracted from Villena *et al*. [61] were fit to a quadratic function using the nls function in R as described in Mordecai *et al*. [30]. The fitted quadratic function is shown, with parameters listed in **S6 Table**.
(TIF)

**S4 Fig. The ratio of $R_0$ values is temperature dependent but consistent across temperatures in 27 ± 2˚C.** (A) Baseline $R_0$ (orange) and Adjusted $R_0^b$ (purple) as temperature varies. No numeric scale is given as raw $R_0$ values depend on parameters, such as population size, that are not temperature dependent and cancel out in the $R_0$ ratio. (B) The increase in $R_0$ with shortened EIP as a function of temperature. Within the temperature range 27 ± 2˚C, the increase in $R_0$ is between 10.1% and 12.1%.
(TIF)

**S5 Fig. Monthly changes in $R_0$ under a multiple blood feeding scenario.** We calculated the monthly changes in $R_0$ for each 5x5 km grid cell by taking the ratio $R_0^b/R_0$ using the mean temperature of each month that has a mean temperature at $27 \pm 2°C$. The restricted data points shown here are used to create the summary maps in **Fig 4**.
(TIF)

**S1 Table. Statistical models.** JMP 14 Pro statistical software was used to construct models for data analysis to account for multiple variables in an experiment. Residual Maximum Likelihood (REML) variance components analysis was used by fitting linear mixed models after cube-root transformation to resemble a normal distribution. The number of blood feeds, dsRNA injection group and their interaction were used as fixed effects and replicate was included as a random effect. Effect test outputs are reported here. Multiple comparisons were calculated using 4 pairwise Student's t tests followed by FDR correction (see **S2 Table**). d pIBM = days post infectious blood meal; #BF = number of blood feeds; FDR = false discovery rate.
(DOCX)

**S2 Table. Post-hoc testing.** Significant differences in oocyst size and oocyst and sporozoite intensity using an FDR of 0.05. See **S1 Table**.
(DOCX)

**S3 Table. Statistical testing for infection prevalence.** GraphPad Prism 8 was used for logistic regression, Fisher's exact and $\chi^2$ tests.
(DOCX)

**S4 Table. Post-hoc testing.** Significant differences in infection prevalence using an FDR of 0.05. See **S3 Table**.
(DOCX)

**S5 Table. Temperature dependent traits fitted with Brière function.** All parameters from Mordecai *et al.* [30]. See references within.
(DOCX)

**S6 Table. Temperature dependent traits fitted with quadratic function.** Parameters for *bc*, *p*, and $p_{EA}$ from Mordecai *et al.* [30]. See references within. Parameters for EFD were fit to data published in Villena *et al.* [61].
(DOCX)

**S1 Data. Raw data plotted in graphical figures.**
(XLSX)

## Acknowledgments

We thank Emily Lund, Kate Thornburg, and Emily Selland for help in mosquito rearing, and members of the Catteruccia laboratory for comments on the manuscript.

## Author Contributions

**Conceptualization:** W. Robert Shaw, Maurice A. Itoe, Kristine Werling, Douglas G. Paton, Caroline O. Buckee, Lauren M. Childs, Flaminia Catteruccia.

**Data curation:** W. Robert Shaw, Inga E. Holmdahl.

**Formal analysis:** W. Robert Shaw, Inga E. Holmdahl, Douglas G. Paton, Caroline O. Buckee, Lauren M. Childs, Flaminia Catteruccia.

**Funding acquisition:** Lauren M. Childs, Flaminia Catteruccia.

**Investigation:** W. Robert Shaw, Inga E. Holmdahl, Maurice A. Itoe, Kristine Werling, Meghan Marquette, Naresh Singh, Lauren M. Childs, Flaminia Catteruccia.

**Methodology:** W. Robert Shaw, Inga E. Holmdahl, Naresh Singh, Lauren M. Childs, Flaminia Catteruccia.

**Project administration:** W. Robert Shaw, Inga E. Holmdahl, Maurice A. Itoe, Kristine Werling, Lauren M. Childs, Flaminia Catteruccia.

**Resources:** W. Robert Shaw, Inga E. Holmdahl, Meghan Marquette, Naresh Singh, Lauren M. Childs, Flaminia Catteruccia.

**Software:** Inga E. Holmdahl, Lauren M. Childs.

**Supervision:** Caroline O. Buckee, Lauren M. Childs, Flaminia Catteruccia.

**Validation:** W. Robert Shaw, Inga E. Holmdahl, Maurice A. Itoe, Kristine Werling, Meghan Marquette, Douglas G. Paton, Lauren M. Childs, Flaminia Catteruccia.

**Visualization:** W. Robert Shaw, Inga E. Holmdahl, Caroline O. Buckee, Lauren M. Childs, Flaminia Catteruccia.

**Writing – original draft:** W. Robert Shaw, Inga E. Holmdahl, Caroline O. Buckee, Lauren M. Childs, Flaminia Catteruccia.

**Writing – review & editing:** W. Robert Shaw, Inga E. Holmdahl, Maurice A. Itoe, Kristine Werling, Meghan Marquette, Douglas G. Paton, Naresh Singh, Caroline O. Buckee, Lauren M. Childs, Flaminia Catteruccia.

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
