## [Decision Letter · Decision Letter 0]

4 Oct 2020

Dear Flaminia,

Thank you very much for submitting your manuscript "Multiple blood feeding in mosquitoes shortens the Plasmodium falciparum incubation period and increases malaria transmission potential" for consideration at PLOS Pathogens. As with all papers reviewed by the journal, your manuscript was reviewed by members of the editorial board and by several independent reviewers. The reviewers appreciated the attention to an important topic. Based on the reviews, we are likely to accept this manuscript for publication, providing that you modify the manuscript according to the review recommendations.

Please take particular care to address the point made by reviewer 1 not to overstate the degree to which your findings undermine existing models (presumably this refers to lines 212 and following), and please also consider discussing your work in the light of Photini Sinnis’s recent work on the non-linear relationship between sporozoite load and transmission success. 

We are sorry it has taken a while to reach a conclusion on your manuscript. It has unfortunately proved difficult on this occasion to secure reviewers covering the different aspects of your manuscript.

Sincerely,

Oliver Billker

Associate Editor

PLOS Pathogens

Kami Kim

Section Editor

PLOS Pathogens

Kasturi Haldar

Editor-in-Chief

PLOS Pathogens

orcid.org/0000-0001-5065-158X

Michael Malim

Editor-in-Chief

PLOS Pathogens

orcid.org/0000-0002-7699-2064

Reviewer Comments (if any, and for reference):

Reviewer's Responses to Questions

**Part I - Summary**

Reviewer #1: This is an interesting paper describing how two blood feeds of Anopheles gambiae mosquitoes fastens the maturation of Plasmodium falciparum sporozoites and hence could lead to more rapid transmission of malaria. They provide a model that goes in this direction and also discuss how current efforts in generating transgenic mosquitoes aimed at mosquito control could be affected by the authors finding. The experiments are well executed and described and I have only comments on the writing / data presentation.

Reviewer #2: This is an excellent and potentially very influential piece by the groups of Childs and Catteruccia. As they have done before, they build upon anecdotal knowledge (here: that a 2nd blood meal accelerates or synchronizes sporozoite development) and provide an evidence base and sketch the potential implications for our understanding of transmission. I have mostly minor comments and requests for clarification.

Reviewer #3: This is an important and interesting study with appealing results. The authors have encapsulated well the research question they raised in this study. The experiment, methods and modelling have been given with clarity, and relevant important assumptions are also clearly explained in detail. The results of simple yet powerful epidemiological model of malaria transmission potential in context of R0 have been supported well with experimental evidences and borrow strength from previous studies. The paper highlights important issues related to the current understanding of malaria transmission potential under simplified assumptions that could lead to an underestimation of R0. The findings of the study are important for current and future modelling of malaria dynamics and for control interventions.

The results of the paper are novel regarding the deepened understanding of how blood feeding behaviour and rate influence the vectorial capacity and likewise – R0 – of malaria. The message it disseminates clear and valuable. That is, the parasite EIP is an important parameter in mosquito models and previous studies have overlooked and do not account for multiple blood feeding scenarios to estimate parasite EIP. The study could set a base for future studies about the similar parasites of different mosquitoes and their variants, and the study will likely motivate many studies onwards to incorporate blood feeding rate in vectorial capacity.

The authors are aware of study limitations and have nicely described by proposing further exploration for future work to other Anopheles-Plasmodium combinations. Clearly the model and results of R0 are sensitivity to the parameters and may change with a set of new parameterization. The ones used are good picks but of course subject to uncertainty over time and space. On the other hand, relating directly to the published study by Mordecai and is of course of value, and makes it easier to the added value of this study.

Overall, I find this is an interesting and important scientific contribution which will have scientific and policy impact.

**Part II – Major Issues: Key Experiments Required for Acceptance**

Reviewer #1: No experiments but I think the following two suggestions will help the authors improve their manuscript:

the authors often report the mean in figures, but the average is not informative in non-normally distributed data, please replace with median in figures and text and state in figure legends.

One major difference in transmission the authors data suggests but the authors do not comment about is that there might be ab increased number of mosquitoes harbouring very high numbers of sporozoites in the two blood meal groups, e.g. see Figure 3B, where there seems to be at least a 5 fold increase in the numbers of highly infected mosquitoes. The authors should indicate this in the results section and discuss it in light of recent laboratory single bite transmission experiments by the Sinnis lab which suggested that there might be.a threshold for efficient transmission. The effects seen with increased number of sporozoites early in salivary glands might indeed be much larger if those few high infecting mosquitoes are indeed the primary driver of transmission, as so often also seen in other infectious diseases.

Reviewer #2: The only element of the work I feel uncomfortable with is the extrapolation of findings to EIR estimates. The most widely used maps for estimating relative transmission intensity are based on parasite prevalence estimates in children (e.g. the maps generated by MAP). Clearly those maps are not going to change based on the current findings and if there are reasons to suspect sporogonic development has not been adequately captured in leading transmission models, clearly some other factors should also be wrong since these models (e.g. the Imperial Model and IDM model) do capture transmission with reasonable precision. I wonder whether this part should be in this manuscript. It underlines the importance of the findings but at the same time leaves me with the impression that the authors to some extent oversell their (definitely relevant) finding.Figure 3C is very strong and convincing and already gives a sense of urgency to the findings, figure 4 is less convincing.

Reviewer #3: None

**Part III – Minor Issues: Editorial and Data Presentation Modifications**

Reviewer #1: Please use sensible digits when reporting numbers, e.g. 2.25 +/- 0.39 should be reported as 2.3 +/- 0.4, 20.7% should read 21% etc

Please indicate in figure legends how many times an infection was performed and that the n corresponds to mosquitoes analysed, e.g. lines 471, 492 etc: replace ‘sample size’ with ‘numbers of mosquitoes analysed from X different experiments’. This info is better placed in the figure legend then in the materials where the authors state that at least 3 infections were performed.

Line 134: does size specify volume or area? Please specify

Figure 3B: maybe indicate on y-axis that these are salivary gland derived or infectious sporozoites?

Figure 3C: maybe indicate on y-axis that these are infectious mosquitoes

Reviewer #2: It might be worth mentioning that, based on the findings by Ponnudurai, also groups that perform experimental infections typically provide a 2nd blood meal to mosquitoes to ensure adequate and timely sporozoite development.

The statement on the average life-span of mosquitoes is crucial for understanding the importance of the findings but may actually be an over-estimate of average mosquito life spans and is based on two older studies. Another old reference (M. W. Service and H. Towson, “The Anopheles vector,” in Essential Malariology, D. A. Warrell and H. M. Gilles, Eds., pp. 59–84, Arnold, London, UK, 2002) actually indicates 10-14 days as average.

The statement in lines 185-186 that oocyst size is a good proxy EIP seems too strong. There is considerable variation in oocyst size and the authors do not present evidence that this variation also contributions to variation in EIP. Is there a minimum oocyst size that, with high 'positive predictive value' predicts completion of EIP within a certain period?

The implications of the findings for mosquitoes with impaired egg development is strong and important.

In the discussion section, I do not fully grasp what the authors intend to say with line 284 'However, the overall impact of additional... feeding frequency'. It seems rather obvious that the prevalence of malaria in a population is important for malaria.

The capelin out of heterogeneities (lines 305-310) needs further clarification.

Figures are clear and relevant.

Reviewer #3: • Since the paper is a mix of experimental and theoretical approaches and will most likely attract readers of both backgrounds, it would be better if the full expression for ‘ds RNA’ (line 120), ‘dsGFP’ (line 121) is also written as it has been consistently done for other terminologies in the paper.

• At line 121, (Fig S1) should explicitly refer to supplementary information

• In modelling section, line 413: ‘on’ should be removed from “g(T) depends on linearly on EIP”.

• Line 429; the accuracy of the temperature considered in model for sub-Saharan Africa could be elaborated.

• temperature data of recent most decades could have been considered too. Is there any patent reason to choose the mean temperature from 1970-2000 (line 429)? It needs some explanation as the relevance of results is tallied with current population (2020) of that region and since temperature is changing rapidly.

• In supplementary material, Table S6, model parameter EFD, for instance represents the eggs laid per adult female ‘aedes albopictus’ per day. If this differs from EFD of malaria mosquitoes it can be replaced.

• I found the change in R0 in relation to the number of months per year of suitable conditions of interest. It appears there is perhaps a relationship there that could be explored further.

PLOS authors have the option to publish the peer review history of their article (what does this mean?). If published, this will include your full peer review and any attached files.

Reviewer #1: No

Reviewer #2: No

Reviewer #3: No
---

## [Editor Report · Decision Letter 1]

8 Nov 2020

Dear Dr Catteruccia,

We are pleased to inform you that your manuscript 'Multiple blood feeding in mosquitoes shortens the Plasmodium falciparum incubation period and increases malaria transmission potential' has been provisionally accepted for publication in PLOS Pathogens.

Best regards,

Oliver Billker

Associate Editor

PLOS Pathogens

Kami Kim

Section Editor

PLOS Pathogens

Kasturi Haldar

Editor-in-Chief

PLOS Pathogens

orcid.org/0000-0001-5065-158X

Michael Malim

Editor-in-Chief

PLOS Pathogens

orcid.org/0000-0002-7699-2064
---

## [Editor Report · Acceptance letter]

7 Dec 2020

Dear Dr Catteruccia,

We are delighted to inform you that your manuscript, "Multiple blood feeding in mosquitoes shortens the *Plasmodium falciparum* incubation period and increases malaria transmission potential," has been formally accepted for publication in PLOS Pathogens.

Best regards,

Kasturi Haldar

Editor-in-Chief

PLOS Pathogens

orcid.org/0000-0001-5065-158X

Michael Malim

Editor-in-Chief

PLOS Pathogens

orcid.org/0000-0002-7699-2064